# Physion: Evaluating Physical Prediction from Vision in Humans and Machines

**Daniel M. Bear**[1,4,*], **Elias Wang**[2,4,*], **Damian Mrowca**[3,*], **Felix Binder**[5,*], **Hsiao-Yu Fish Tung**[1,7],
**R.T. Pramod**[7], **Cameron Holdaway**[6], **Sirui Tao**[6], **Kevin Smith**[7], **Fan-Yun Sun**[3], **Li Fei-Fei**[3],
**Nancy Kanwisher**[7], **Joshua B. Tenenbaum**[7], **Daniel L. K. Yamins**[1,3,4,**], **and Judith Fan**[6,**]

Department of Psychology[1], Electrical Engineering[2], and Computer Science[3], and
Wu Tsai Neurosciences Institute[4], Stanford, CA 94305
Department of Cognitive Science[5], and Psychology[6], UC San Diego, CA 92093
Department of Brain and Cognitive Sciences and CBMM[7], MIT, Cambridge, MA 02139

{`dbear`, `eliwang`, `mrowca`}@stanford.edu, `fbinder`@ucsd.edu

## Abstract

While current vision algorithms excel at many challenging tasks, it is unclear how well they understand the physical dynamics of real-world environments. Here we introduce **Physion**, a dataset and benchmark for rigorously evaluating the ability to predict how physical scenarios will evolve over time. Our dataset features realistic simulations of a wide range of physical phenomena, including rigid and soft-body collisions, stable multi-object configurations, rolling, sliding, and projectile motion, thus providing a more comprehensive challenge than previous benchmarks. We used **Physion** to benchmark a suite of models varying in their architecture, learning objective, input-output structure, and training data. In parallel, we obtained precise measurements of human prediction behavior on the same set of scenarios, allowing us to directly evaluate how well any model could approximate human behavior. We found that vision algorithms that learn object-centric representations generally outperform those that do not, yet still fall far short of human performance. On the other hand, graph neural networks with direct access to physical state information both perform substantially better and make predictions that are more similar to those made by humans. These results suggest that extracting physical representations of scenes is the main bottleneck to achieving human-level and human-like physical understanding in vision algorithms. We have publicly released all data and code to facilitate the use of **Physion** to benchmark additional models in a fully reproducible manner, enabling systematic evaluation of progress towards vision algorithms that understand physical environments as robustly as people do.

## 1 Introduction

Vision algorithms that understand the physical dynamics of real-world environments are key to progress in AI. In many settings, it is critical to be able to anticipate when an object is about to roll into the road, fall off the table, or collapse under excess weight. Moreover, for robots and other autonomous systems to interact safely and effectively with their environment they must be able to accurately predict the physical consequences of their actions.

---

[*]/[**]Equal contribution

35th Conference on Neural Information Processing Systems (NeurIPS 2021), Sydney, Australia.

## 1.1 Establishing Common Standards for Evaluating Physical Understanding

Despite recent progress in computer vision and machine learning, it remains unclear whether any vision algorithms meet this bar of everyday physical understanding. This is because previously developed algorithms have been evaluated against disparate standards — some prioritizing accurate prediction of every detail of a scenario's dynamics and others that only require predictions about a specific type of event.

The first set of standards has generally been used to evaluate algorithms that operate on unstructured video inputs, such as in robotics [20]. These algorithms typically aim for fine-grained prediction of upcoming video frames or simulation of the trajectories of individual particles. However, only algorithms with near-perfect knowledge of the world's physical state – like Laplace's Demon – could hope to predict how a complete set of events will unfold. This explains why models of this kind have sufficed in less varied visual environments, but underfit on more diverse scenarios [17, 39]. Though recent efforts to scale these algorithms have led to improvements in the quality of predicted video outputs [65, 68], it remains to be seen whether their learned representations embody more general *physical* knowledge.

The second set of standards has been used to probe qualitative understanding of physical concepts, especially in cognitive and developmental psychology [4, 60, 15]. Much of this work has focused on measuring and modeling human judgments about discrete events, such as whether a tower of blocks will fall over or whether an object will reemerge from behind an occluder [10, 8, 5]. Findings from this literature suggest that humans simulate dynamics over more abstract representations of visual scenes to generate reliable predictions at the relevant level of granularity [49, 57]. However, existing models that instantiate such simulations typically require require structured input data (e.g., object segmentations) that may not be readily available in real-world situations [35, 32]. Moreover, the abstractions that are appropriate for one task may not work well in more general settings [64, 67, 43].

A key challenge in developing improved visual models of physical understanding is thus to establish common standards by which to evaluate them. Here we propose such a standard that both combines elements of previous approaches and goes beyond them: we require models to operate on highly varied and unstructured visual inputs to generate event-based predictions about a wide variety of physical phenomena. By contrast with prior efforts to evaluate vision algorithms, our proposed standard argues for the importance of considering a wider variety of physical scenarios and the ability to compare model predictions directly with human judgments. By contrast with prior efforts to model human physical understanding, our approach embraces the challenge of generating predictions about key events from realistic visual inputs.

## 1.2 Desiderata for a Generalized Physical Understanding Benchmark

We envision our generalized physical understanding benchmark as combining two key components: first, a dataset containing visually realistic and varied examples of a wide variety of physical phenomena; and second, a generic evaluation protocol that probes physical understanding in a way that is agnostic to model architecture and training regime.

**Dataset.** While there are several existing datasets that probe physical understanding to some extent, each of them fall short on at least one key dimension. Some datasets contain realistic visual scenes but do not adequately probe understanding of object dynamics [17]. Other datasets feature realistic scenarios with challenging object dynamics, but consider only a narrow set of physical phenomena, such as whether a tower of blocks will fall [29] or whether a viewed object's trajectory violates basic physical laws [49, 46, 57]. Other datasets featuring a greater diversity of physical phenomena are designed in simplified 2D environments that may not generalize to real-world 3D environments [6].

**Evaluation protocol.** In order to test a wide variety of models in a consistent manner, many commonly used evaluations will not suffice. For example, evaluations that query the exact trajectories of specific objects [9, 16] are not well posed for models that do not extract explicit object representations. Conversely, evaluations that depend on image matching or visual realism-based metrics [21, 69, 17, 68] are not straightforward to apply to models that do not re-render images. A more promising approach to measuring physical understanding in a model-agnostic manner may instead take inspiration from prior work investigating human physical prediction ability [10, 51, 8], which does not assume that the trajectories of all objects in a scene are represented with perfect fidelity.

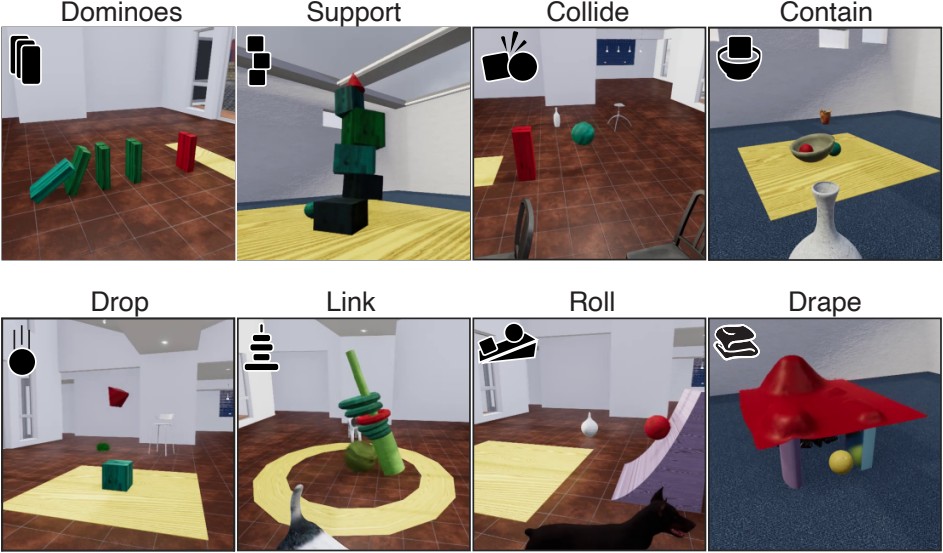

Figure 1: Example frames from the eight **Physion** scenarios. Red object is agent; yellow is patient.

## 1.3 Physion: A Dataset and Benchmark for Physical Understanding

In recognition of the above desiderata, we developed **Physion**, a new physical understanding dataset and benchmark. Our dataset contains a wide variety of visually realistic examples of familiar physical phenomena, including: collisions between multiple objects; object-object interactions such as support, containment, and attachment; projectile, rolling, and sliding motion that depends on object geometry; and the behavior of soft materials like cloth. For each of these eight scenario types (1), we operationalize physical understanding using the **object contact prediction (OCP) task**, which prompts agents to predict whether two cued objects will come into contact as a scene unfolds.

## 1.4 Using Physion to Benchmark Human and Model Physical Understanding

In addition to the dataset, we introduce a unified evaluation protocol for directly comparing model and human behavior. Approximating human physical understanding from vision is a natural target for AI systems for two key reasons: first, humans have already demonstrated their ability to competently navigate a wide variety of real-world physical environments; and second, it is important for AI systems to anticipate how humans understand their physical surroundings in order to co-exist safely with people in these environments. Towards this end, our paper conducts systematic comparison between humans and several state-of-the-art models on the same physical scenarios.

Our experiments feature a wide range of models that vary in their architecture, learning objective, input-output structure, and training regime. Specifically, we include vision models that make pixel-level predictions via fully convolutional architectures, [23, 1, 36, 21, 35, 70, 40, 41, 66, 30, 34, 54, 27]; those that either explicitly learn object-centric representations of scenes [64, 33, 19, 27, 50] or are encouraged to learn about objects via supervised training [56, 62]; and physics dynamics models that operate on object- or particle-graph representations provided as input [16, 9, 37, 8, 61, 52, 11, 42, 2, 57, 69, 47].

Models that perform physical simulation on a graph-like latent state are especially attractive candidates for approximating human prediction behavior, based on prior work that has found that *non*-machine learning algorithms that add noise to a hard-coded simulator accurately capture human judgments in several different physical scenarios [10, 51, 7, 13]. Consistent with these results, recurrent graph neural networks supervised on physical simulator states can learn to accurately predict full object trajectories [42, 37, 38, 53]. However, these models have not been tested for their ability to generalize across diverse, multi-object scenarios, and they require such detailed physical input and trajectory supervision that they have so far not been useful in cases where only realistic sensory observations are available.

Among models that take visual input, object-centric predictors in some cases make more accurate predictions than those that simulate scene dynamics in pixel space [64, 47, 19]; however, these comparisons have only been done in reduced environments with few distinct physical phenomena, so it is not known whether this result holds in more realistic settings. Indeed, models that make pixel-level predictions are standard in robotics applications [34, 68] due to the longstanding difficulty of inferring accurate object-centric representations from raw video data without supervision, despite recent progress [14, 64, 12].

### 1.5 Summary of Key Findings

By assessing many models on the same challenging physical understanding task, our experiments address previously unresolved questions concerning the roles of model architecture, dataset, and training protocols in achieving robust and human-like physical understanding. We found that no current vision algorithms achieve human-level performance in predicting the outcomes of **Physion** scenes. Vision algorithms encouraged to learn object-centric representations generally outperform those that do not, yet still fall far short of human performance. On the other hand, particle-based models with direct access to physical state information both perform substantially better and make predictions that are more similar to those made by humans. Taken together, these results suggest that extracting physical representations of visual scenes is the key bottleneck to achieving human-level and human-like physical understanding in vision algorithms.

### 1.6 Our Vision for Physion

Our initial public release of **Physion** includes large, labeled training and test datasets for each scenario, as well as code for for generating additional training data. As such, one potential way to use **Physion** is to train additional models directly on the OCP task for one or more of the scenarios, yielding, for example, a model that excels at predicting whether block towers will fall. However, the primary use case we have in mind for **Physion** is to test how well pretrained models transfer to challenging physical understanding tasks, analogous to how humans make predictions about **Physion** videos without extensive training on the OCP task. Towards this end, we have shared code to facilitate the use of the **Physion** test dataset to benchmark additional models in a fully reproducible manner, enabling systematic evaluation of progress towards vision algorithms that understand physical environments as robustly as people do.

## 2 Methods

### 2.1 Benchmark Design

We used the ThreeDWorld simulator (TDW), a Unity3D-based environment [24], to create eight physical scenarios out of simple objects that incorporate diverse physical phenomena (Fig. 1):

1. **Dominoes** – sequences of collisions that depend on the arrangement and poses of objects
2. **Support** – stacks of objects that may fall over, depending on their shapes and arrangement
3. **Collide** – pairs of objects that may collide, depending on their placement and trajectories
4. **Contain** – container-like objects that may constrain other objects by virtue of their shapes
5. **Drop** – objects falling and bouncing under the force of gravity
6. **Link** – objects restricted in their motion because they are attached to other objects
7. **Roll** – objects that move across a surface either by rolling or sliding
8. **Drape** – cloth draping over other objects by virtue of their shape and the cloth's material.

In each scenario, contact between agent and patient serves as a non-verbal indicator of some physical higher-order variable – whether a tower fell over, a bowl contained a ball, a torus was attached to a post – whose prediction should require understanding of the relevant physical phenomena. Together, these scenarios cover much of the space of physical dynamics possible through simple rigid- and soft-body interactions; additional scenarios will be developed to include other material types (e.g., "squishy" objects, fluids) and complex interactions (e.g. multi-part, jointed objects.)

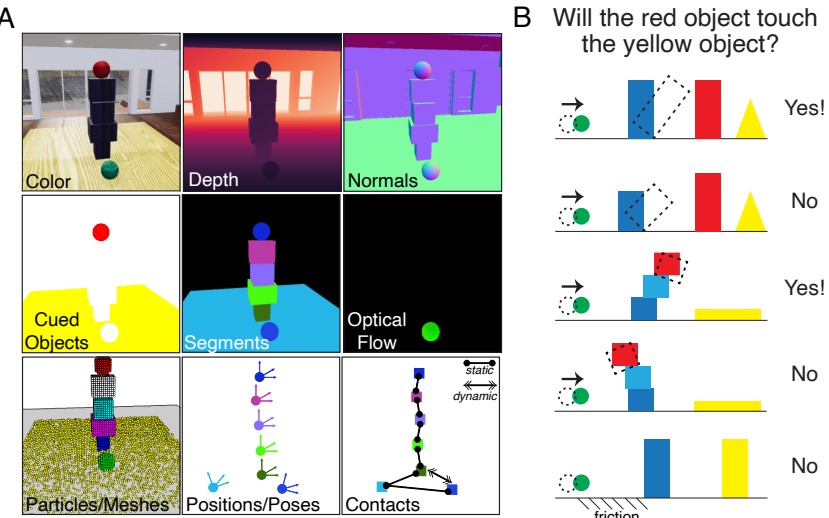

Figure 2: Stimulus attributes and task design. (**A**) Output of TDW for an example frame of a stimulus movie. (**B**) A schematic of the OCP task: humans and models must predict whether the *agent* object (red) will contact the *patient* (yellow), given the initial setup and the motion of the *probe* (green).

## 2.2 Stimulus Generation and Task Design

We constructed scenes out of basic "toy blocks" to avoid confounds from knowledge of object configurations that are common in the real world (e.g., cups typically appearing on tables); rather, accurate predictions should require judgments about objects' physical properties, relationships, and dynamics. To increase physical variability *within* each scenario, we identified multiple configurations of simulator parameters that lead to different types of physical dynamics. Configurations specify distributions of initial scene variables, such as the positions of objects; they also introduce substantial visual variation that does not affect the physical outcome of the scene, including variation in camera position and pose, object colors and textures, the choice of "distractor" object models that do not participate in scene dynamics, and the appearance of the background. Training and testing stimuli were generated by randomly sampling initial conditions and scene properties according to each configuration, then running the simulation until all objects came to rest. Additional stimuli can be generated by sampling further from our configurations or by creating new ones. Examples of stimuli from each scenario can be found in the Supplement.

Each stimulus is a 5-10 second movie rendered at 30 frames per second. For model training and evaluation we also supply the full output of the TDW simulation (Fig. 2A), which includes: 1.) *visual data per frame*: color image, depth map, surface normal vector map, object segmentation mask, and optical flow map; 2.) *physical state data per frame*: object centroids, poses, velocities, surface meshes (which can be converted to particles), and the locations and normal vectors for object-object or object-environment collisions; 3.) *stimulus-level labels and metadata*: the model names, scales, and colors of each object; the intrinsic and extrinsic camera matrices; segmentation masks for the agent and patient object and object contact indicators; the times and vectors of any externally applied forces; and scenario-specific parameters, such as the number of blocks in a tower. All stimuli from all eight scenarios share a common OCP task structure (Fig. 2B): there is always one object designated the *agent* and one object designated the *patient*, and most scenes have a *probe* object whose initial motion sets off a chain of physical events. Models and people are asked to predict whether the agent and patient object will come into contact by the time all objects come to rest. We generated trials for human testing by sampling from scenario-specific configurations until we had 150 testing stimuli per scenario with an equal proportion of contact and no-contact outcomes.

## 2.3 Testing Humans on the Physics Prediction Benchmark

**Participants.** 800 participants (100 per scenario; 447 female, 343 male, 7 declined to state; all native English speakers) were recruited from Prolific and paid $4.00 for their participation. Each was shown all 150 stimuli from a single scenario. Data from 112 participants were excluded for not

meeting our preregistered inclusion criterion for accurate and consistent responses on attention-check trials (see Supplement). Our preregistered analysis plan is stored under version control in our GitHub repository. These studies were conducted in accordance with the UC San Diego and Stanford IRBs.

**Task procedure.** The structure of our task is shown in Fig. 3A. Each trial began with a fixation cross, which was shown for a randomly sampled time between 500ms and 1500ms. To indicate which of the objects shown was the agent and patient object, participants were then shown the first frame of the video for 2000ms. During this time, the agent and patient objects were overlaid in red and yellow respectively. The overlay flashed on and off with a frequency of 2Hz. After this, the first 1500ms of the stimulus were played. After 1500ms, the stimulus was removed and the response buttons were enabled. Participants proceeded to the next trial after they made a prediction by selecting either "YES" (the agent and patient would touch) or "NO" (they would not). The order of the buttons was randomized between participants. Before the main task, participants were familiarized with 10 trials that were presented similarly to the test trials, except (a) the full stimulus movie and accuracy feedback was presented after participants indicated their prediction, and (b) all trials were created from basic templates without occluding and distracting objects. Familiarization trials were always presented in the same order. After the test trials were completed, basic demographics were collected from participants. Finally, participants were informed of their overall accuracy.

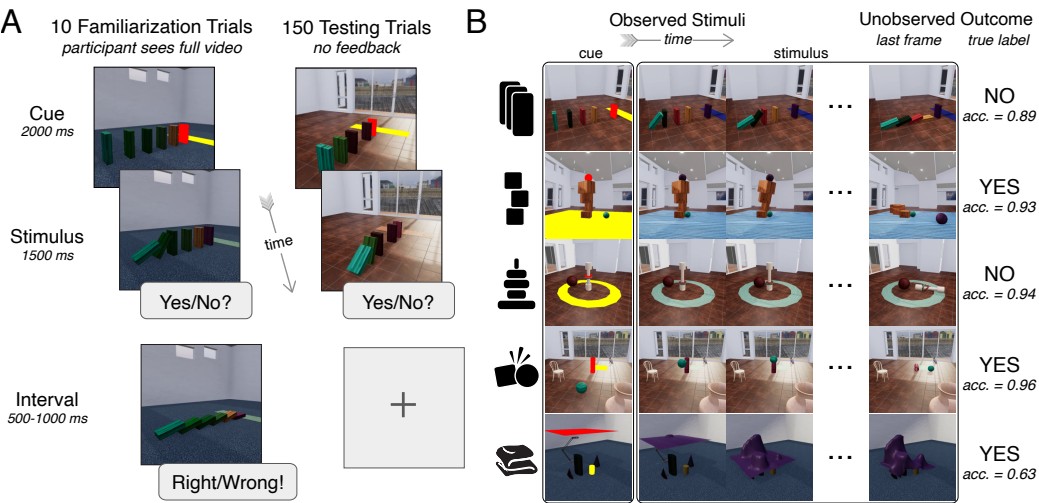

Figure 3: Human task. (A) Trial structure for the familiarization trials (*left*) and test trials (*right*) indicating the Cue, Stimulus, and Inter-trial periods. (B) Example stimuli (rows) including the last frame (not shown during the experiment). Last column indicates the outcome and human accuracy.

## 2.4 Benchmarking Computer Vision and Physical Dynamics Models

We developed a standard procedure for training machine learning models and evaluating any image- or physical state-computable algorithm on the benchmark. Let $\{X_t\}_{i=1}^{N_{test}}$ be the set of $N_{test}$ testing stimuli for a single benchmark scenario, where $\{X_t\}_i$ denotes the ordered set of RGB images that constitutes the full movie of stimulus $i$ and $\{X_{1:t_{vis}}\}$ the truncated movie shown to participants. Further let $\mathcal{O}_i := \{o_1, o_2, ..., o_{K_i}\}$ denote unique IDs for each of the $K$ objects being simulated in this stimulus. Doing the OCP task can be formalized as making a binary contact prediction by applying to the testing stimuli a function $\mathcal{F}_\Theta : (\{X_{1:t_{vis}}\}, o_a, o_p) \mapsto P(contact)$, where $o_a$ is the agent object, $o_p$ is the patient, and $P(contact)$ is the predicted probability that they will come into contact. For people, feedback on only ten familiarization trials is sufficient to learn such a function. To adapt any image-computable model to the OCP task, we apply the following procedure. First, we assume that a model can be decomposed into a *visual encoder* that maps an input movie to a state-vector representation of each frame; a *dynamics predictor* that predicts unseen future states from the "observed" state vector; and a *task adaptor* that produces a trial-level response $P(contact)$ from the concatenation of the observed and predicted state vectors (Fig. 4). In general, models will include only a visual encoder and possibly a dynamics predictor in their original design; the task

adaptor is added and fit as part of our model evaluation pipeline, where it removes the need for the explicit trial-level cueing with superimposed object masks (see below.)

**Testing, Readout Fitting, and Training sets.** Each **Physion** scenario consists of three stimulus sets: *Testing*, *Readout Fitting*, and *Training*. The *Testing* stimuli are identical to the 150 trials per scenario shown to humans, except that the agent and patient objects are permanently colored red and yellow (Fig. 1) instead of being indicated by red and yellow masks on the first frame (Fig. 3). This difference allows models to be tested on RGB movie stimuli alone, without providing segmentation masks that most computer vision model architectures are not designed to handle as inputs. Each trial in the *Testing* sets includes the ground truth label of whether it ends in agent-patient contact and the responses of >100 human participants. We also provide the *Human Testing* stimuli with red and yellow cueing masks rather than permanently colored objects.

Each scenario's *Readout Fitting* set consists of 1000 stimuli generated from the same configurations as the *Testing* stimuli, such that the two sets have the same visual and physical statistics. The *Readout Fitting* stimuli are for fitting a OCP task-specific adaptor to each model. In designing **Physion**, we did not want to restrict testing only to models optimized directly to do the OCP prediction task. Thus, during evaluation we freeze the parameters of a pretrained model and fit a generalized linear model, the task adaptor, on various subsets of model features (see below). The *Readout Fitting* stimuli are the training set for this fitting procedure, with the ground truth object contact labels acting as supervision. This allows the task adaptor to generalize to the *Testing* stimuli.

Finally, each scenario's *Training* set includes 2000 movies generated from the same configurations as the *Testing* and *Readout Fitting* stimuli, but with no visual features indicating agent and patient objects. The purpose of the *Training* sets is to let models learn or fine-tune representations of physical dynamics in a way that is agnostic to any particular readout task: a model partly or entirely trained on a "non-physics" task like object categorization might nevertheless acquire a human-like representation of the physical world, which **Physion** should reveal via transfer learning. During training models see movie clips sampled from the entirety of each *Training* stimulus, not just the initial portion seen during readout fitting and testing, and they do not receive ground truth OCP labels.

The procedure for training a given model depends on its original architecture and optimization procedure. For models that take multi-frame inputs and include both a visual encoder and a dynamics predictor in their architecture, we train the full model end-to-end on the *Training* sets. For models that include only a visual encoder pretrained on another dataset and task (such as ImageNet), we add an RNN dynamics model that predicts future encoder outputs from the "observed" encoder outputs on an input frame sequence; the training loss is the mean squared error between each predicted output and the matching observed output, which optimizes the dynamics model. For these models, we train two versions: one in which the pretrained encoder parameters are fine-tuned and one in which they are frozen. See **Model Comparison** below and the Supplement for further details.

**Model comparison.** To get an overview of how current physical prediction algorithms compare to humans, we tested models from four classes (see Supplement for model details):

1. fully unsupervised, joint encoder-dynamics predictors trained only on the benchmark scenario data: **SVG** [18], **OP3** [64], **CSWM** [33];

2. encoder-dynamics models supervised on ground truth object data: **RPIN** [47];

3. visual encoders *pretrained with supervision on ImageNet* and extended with RNN dynamics predictors, which are trained in an *unsupervised* way on the benchmark scenario data: **pVGG-mlp/lstm** [56], **pDeIT-mlp/lstm** [62];

4. particle-relation graph neural network dynamics predictors that take the ground truth simulator state as input and have no visual encoder (i.e. assume perfect observability of physical dynamics): **GNS** [53], **GNS-RANSAC**, **DPI** [37].

**Training protocols.** We tested models given three types of training (Fig. 4, left): *all*, training on all scenarios' training sets concurrently; *all-but*, training on all scenarios except the one the model would be tested on; and **only**, training on only the scenario type the model would be tested on. We consider the *all* protocol to be the best test of physical understanding, since it produces a model that is not specialized to a specific scenario. Differences between *all* and *all-but* or *only* indicate how well a model can generalize across scenarios or overfit to a single scenario, respectively.

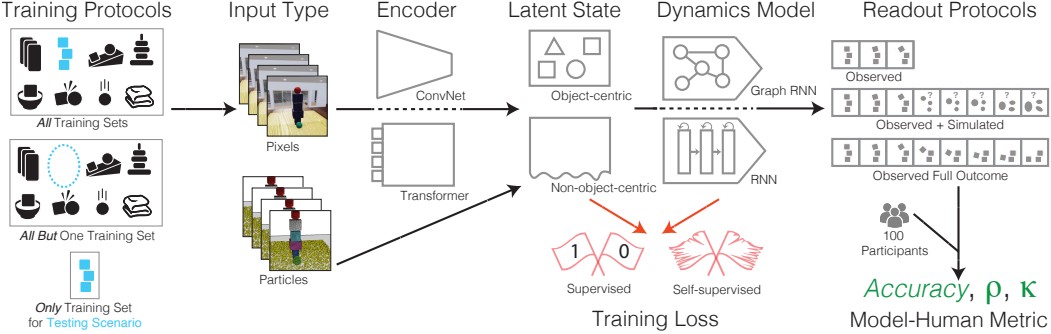

Figure 4: The model benchmarking pipeline including training, architecture, and readout variants.

**Testing protocols.** We fit logistic regression models as OCP task adaptors with three protocols (Fig. 4, right): *observed*, in which adaptors are fit only to the features produced by showing the human stimulus (first $t_{vis}$ frames, equivalent to 1.5 seconds) to the model's visual encoder; *observed+simulated*, which uses the *observed* features concatenated with the "simulated" features output by the model's dynamics predictor; and *full*, which uses the features produced from showing the entire movie (not just the testing stimulus portion) to the visual encoder. Outputs from the *full* protocol cannot be directly compared to human data, since they represent a model's performance on a detection (rather than prediction) task; however, we use them to assess how well physical information is encoded in a model's visual features (see Experiments.) We compare a model's outputs to human responses on each scenario's testing stimuli with three standard metrics (Fig. 4, right): overall accuracy, Pearson correlation between model and average human responses across stimuli, and Cohen's $\kappa$, a measure of how much a model's binary predictions resemble a single human's, averaged across participants. For all three metrics, we assess how close models are to the "human zone" – the empirical distribution of each statistic across humans or human-human pairs.

## 3    Results and Discussion

**Human behavior is reliable, with substantially above-chance performance.** Human performance was substantially above chance across all eight scenarios (proportion correct = 0.71, t=27.5, p<$10^{-7}$, Fig. 5A), though there was variation in performance across scenarios (e.g., higher accuracy on **Roll** than **Link** or **Drape**). Moreover, the "human zones" for all metrics (raw performance, correlation-to-average, and Cohen's $\kappa$) were tight and far from chance (gray horizontal bars in Fig. 5A-E), showing that the human response patterns were highly reliable at our data collection scale and thus provide a strong empirical test for discriminating between models. Interestingly, each scenario included some stimuli on which the participant population scored significantly *below* chance (Fig. S1). Many of these "adversarial" stimuli had objects teetering on the brink of falling over or other unlikely events occurring after the observed portion of the movie. People may have accurately judged that most scenes *similar to* the observed stimulus would have one outcome, unaware that the other outcome actually occurred due to a physical fluke. This pattern of reliable errors is especially useful for comparing models with humans: if stimuli that fool people do not fool a model, it would suggest that the model draws on different information or uses a non-human strategy for making predictions.

**Particle-based models approach human performance levels, with strong generalization.** Models that received ground-truth TDW object particles as input and supervision (**GNS**, **GNS-RANSAC**, **DPI**) matched human accuracy on many scenarios, with the object-centric **DPI** reaching across-scenario human performance levels (Fig. 5A). These data are consistent with findings that probabilistic physical simulations can account for behavioral judgments on single scenarios that resemble ours [10, 51, 7, 13]. However, our results go beyond prior work in several ways. First, these three models are graph neural networks that *learn* to simulate physical scenes rather than assuming access to a "noisy" version of ground truth dynamics directly provided by the physics engine. Second, the models here performed well above chance when trained with the *all* and *all-but* protocols, not just when they were fit to single scenario types (*only*) as in the work where they were developed [37, 53] (Fig.

5A,E). These results imply that a single graph neural network can learn to make human-level physical predictions across a diverse set of physical scenarios.

**Vision-based models substantially underperform humans, but object-related training may help.** Particle input models have an enormous advantage over both humans and vision models: they operate on ground truth physical information that, in the real world, can never be observed directly, such as the 3D positions, poses, trajectories, and fine-scale shapes of all objects and their occluded surfaces. Whereas humans overcome these limits, none of the vision algorithms here came close to performing at human levels (Fig. 5A). Not all vision models were equally far off, though: among those whose encoders and dynamics simulators were fully unsupervised, **SVG**, a model with only convolutional latent states, performed nearly at chance levels; **OP3**, an object-centric model trained by rendering pixel-level future predictions (b=0.06, t=7.6, p<$10^{-11}$), performed marginally better; while **CSWM**, a model with contrastively-learned object-centric latent states, significantly outperformed both **SVG** and **OP3**. Interestingly, the *supervised* object-centric model **RPIN** was only more accurate than **CSWM** when trained with the *all-but* and *only* protocols, but not the *all* protocol (b=0.035, t=3.7, p<$10^{-3}$, Fig. 5A,E); further experiments are needed to test whether exactly matching the architectures of the two models would reveal a larger effect of ground truth supervision. Together, these results suggest that learning better object-centric representations from realistic, unlabeled video should be a core aim of visual prediction approaches.

The models with *ImageNet-pretrained* ConvNet encoders (**pVGG-mlp/lstm**) significantly outperformed the best fully TDW-trained models (**CSWM**, **RPIN**, b=0.015, t=2.9, p<0.01), and were themselves outperformed by models with ImageNet-pretrained Transformer encoders (**pDeIT-mlp/lstm**, b=0.067, t=16.5, p<$10^{-15}$). This suggests that (supervised) ImageNet pretraining and a better (and perhaps, more "object-aware"-attention driven) encoder architecture produce visual features that are better for physical prediction even *without* learning to explicitly simulate the future. Together these results highlight the importance of learning a "good" visual representation; vision algorithms may benefit from training their encoders on separate tasks and data before learning dynamics predictors.

**Error-pattern consistency is strongly correlated with performance, but a substantial gap remains.** A striking feature of our results is that error-pattern consistency as measured either by correlation-to-average human or Cohen's $\kappa$ (Fig. 5B-C) is itself strongly correlated with absolute model performance. In other words, models that performed better on the prediction task also made errors that were more like those made by humans, strongly analogous to the situation with core visual object recognition [48]. This result suggests, albeit weakly, that human behavior has been highly optimized either directly for a prediction task like that measured in this paper, or for something highly correlated with it. However, none of the models fully reached the "human zone" in which their outputs would be statistically indistinguishable from a person's. This means that even the particle-based models can be improved to better match the judgments people make, including errors; prior work suggests that adding noise to these models could better recapitulate human mental "simulation" [10, 8, 58]. Consistent with this possibility, we found that the particle-based models' predictions were uncorrelated with human predictions on the "adversarial" stimuli, many of which would have opposite outcomes if their initial conditions were slightly different (Fig. S2). Adding noise to the models' forward dynamics might therefore mimic how humans make predictions about *probable* outcomes, rather than simulating dynamics so precisely that they capture even rare flukes.

**What have vision-based models actually learned?** Vision model predictions from the *observed+simulated* readout protocol were, overall, no better than predictions from the *observed* protocol (p=0.53, Fig. 5D). This implies that none of the visual dynamics models learned to "simulate" anything about the scenes that helped on the OCP task (though dynamics predictions during end-to-end training could have usefully shaped the encoder representations.) Rather, any above-chance performance for the vision models was likely due to having visual features that could discriminate some trial outcomes from cues in the initial movie segment. Understanding what makes these visual features useful is the subject of ongoing work: they could be an example of non-causal "shortcut learning" [26] or they could encode important physical properties like object position, shape, and contact relationships. The latter possibility is further supported by two observations. First, the *full* readout protocol yielded significantly higher accuracy for the vision models (b=0.094, t=12.0, p<$10^{-15}$, Fig. 5D), indicating that the learned visual features *are* useful for object contact *detection*. Thus, the best visual features carry some information about the observed objects' spatial relationships, and their relative failures in the *observed* protocol can be fairly said to be these models' lack of physical "understanding." Second, the ImageNet-pretrained models benefited the most from observing the

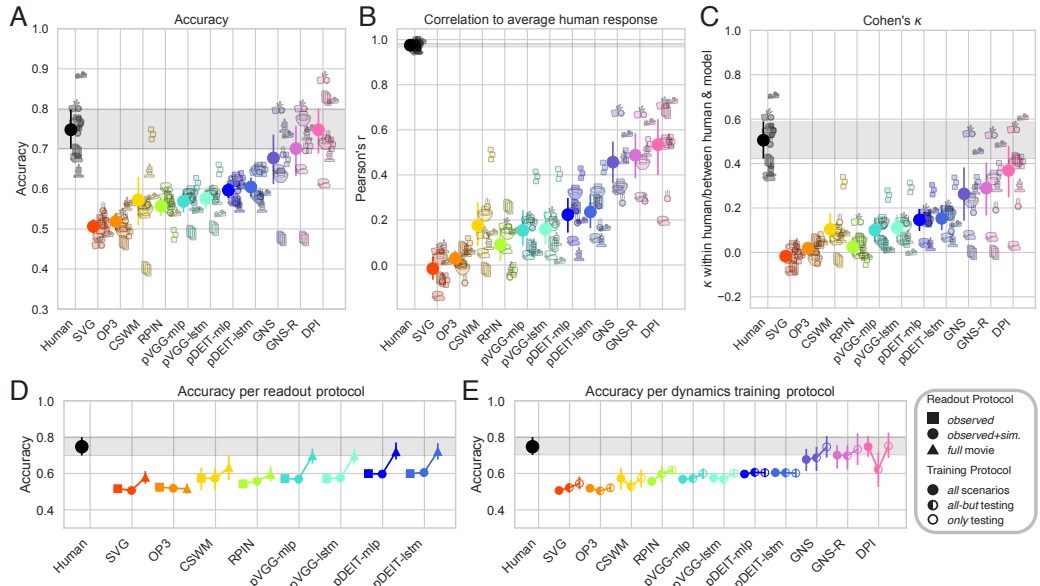

Figure 5: Comparisons between humans and models. First row: the *all*-scenarios trained, *observed+simulated*-readout task accuracy (**A**), Pearson correlation between model output and average human response (**B**), and Cohen's $\kappa$ (**C**) for each model on each scenario, indicated by its icon. Black icons and the gray zones (2.5th-97.5th percentile) show human performance, mean correlation between split halves of participants, and mean human-human Cohen's $\kappa$, respectively. Second row: accuracy of models across the three readout (**D**) and training (**E**) protocols; note that particle-input models have only the *observed+simulated* readout protocol, as predictions are made based solely on whether two objects came within a threshold distance at the end of the predicted dynamics.

full movie, raising the possibility that their pretraining actually captured *more* physically-relevant information than object-centric learning on TDW. Untangling this will require finer-scale comparison between encoder architectures, training datasets, and various supervised and self-supervised losses.

**Having sufficient variability across physical scenarios promotes strong generalization.** Compared to models trained concurrently on *all* scenarios, vision-based models performed only slightly better when they were trained with the *only* protocol (b=0.21, t=4.4, p<$10^{-4}$), and not significantly worse when they were trained with the *all-but* protocol (b=0.009, t=1.9, p=0.057, Fig. 5E). Differences between protocols were larger for particle-based models, but nonetheless small relative to overall performance levels. These results strongly suggest that performance assessments are robust to the specific choices of scenarios we made. This makes sense because the diverse physical phenomena in our everyday environment result from a smaller set of underlying laws. Our results thus quantitatively support the qualitative picture in which an intuitive, approximate understanding of those laws gives rise to humans' outstanding ability to predict and generalize to previously unseen physical phenomena from an early age [60, 15, 5, 49]. However, we do find that models trained on any single scenario do not generalize well to most other scenarios (Fig. S5), suggesting that having substantial diversity of observations is critical for learning general physical forward predictors. It will be important, then, to develop additional testing scenarios that incorporate physical phenomena *not* covered here, such as "squishy" and fluid materials, the dynamics of jointed multi-part objects, and much larger ranges of mass, friction, density, and other physical parameters. We thus hope that our benchmark can be used to drive the development of algorithms with a more general, human-like ability to predict how key events will unfold and to anticipate the physical consequences of their own actions in the real world.

## Acknowledgments

D.M.B. is supported by a Wu Tsai Interdisciplinary Scholarship and is a Biogen Fellow of the Life Sciences Research Foundation. C.H. is supported by a Department of Defense National Defense Science and Engineering Graduate Fellowship. H.F.T., K.A.S, R.T.P., N.K., and J.B.T are supported by National Science Foundation Science Technology Center Award CCF-1231216 and Office of Naval Research Multidisciplinary University Research Initiative (ONR MURI) N00014-13-1-0333; K.A.S. and J.B.T. are supported by research grants from ONR, Honda, and Mitsubishi Electric. D.L.K.Y is supported by the McDonnell Foundation (Understanding Human Cognition Award Grant No. 220020469), the Simons Foundation (Collaboration on the Global Brain Grant No. 543061), the Sloan Foundation (Fellowship FG-2018-10963), the National Science Foundation (RI 1703161 and CAREER Award 1844724), and hardware donations from the NVIDIA Corporation. K.A.S., J.B.T., and D.L.K.Y. are supported by the DARPA Machine Common Sense program. J.E.F. is supported by NSF CAREER Award 2047191 and the ONR Science of Autonomy Program. This work was funded in part by the HAI-Google Cloud Credits Grant Program and the IBM-Watson AI Lab. We thank Seth Alter and Jeremy Schwartz for their help on working with the ThreeDWorld simulator.

## Broader Impact

There are few aspects of everyday life that are not informed by our intuitive physical understanding of the world: moving and doing tasks around the home, operating motor vehicles, and keeping one's body out of harm's way are just a few of the broad behavioral categories that involve making predictions of how objects in the world will behave and respond to our actions. Although there may be ways for algorithms to safely and effectively perform specific tasks without general, human-like understanding of the physical world, this remains a wide open question in many of the areas where AI is rapidly being deployed: self-driving vehicles, robotics, and other systems that involve a "perceive-predict-act" feedback loop. As such, we think the **Physion** benchmark is an important step toward actually measuring whether a given algorithm *does* perceive visual scenes and make physical predictions the way people do. If it turns out that this is critical for achieving safe, high performance in some real-world domain, our benchmark (or its successors) could be used to screen for algorithms more likely to behave like people and to diagnose failures, e.g. by breaking them down into problems making predictions about particular physical phenomena. Moreover our results, though representing only an initial survey of existing algorithms, *do* suggest that models with more explicit physical representations of the world, including the grouping of scene elements into objects, are better equipped to make accurate predictions; they therefore begin to address longstanding questions in AI about whether some sort of "symbolic" representation, inspired by cognitive science, is necessary for an algorithm to accurately predict and generalize to new situations. Though such representations have fallen out of favor in large-scale visual categorization tasks, the fact that they outperform their less or non-symbolic counterparts on the **Physion** tasks raises the intriguing possibility that two broad types of understanding, "semantic" and "physical", may benefit from different algorithm architectures and learning principles. If this is the case, we should reevaluate popular claims that symbolic representations and "interpretable" algorithms are red herrings for making progress in AI.

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
