# OpenReview forum: "Physion: Evaluating Physical Prediction from Vision in Humans and Machines"
_NeurIPS.cc/2021/Track/Datasets_and_Benchmarks/Round1 — NeurIPS 2021 Datasets and Benchmarks Track (Round 1)_

### Official Review · Reviewer_g6xh · 2021-06-21
**An interesting benchmark for evaluation of phyiscal prediction from vision in humans and machines**

**Rating:** 6
**Confidence:** 4
**Correctness:** Please refer to the weakness section.
**Clarity:** It is well written.

**Strengths:**

1. The benchmark is valuable for evaluating machines' intuitive physical understanding of the world, which is of great importance for human cognitive understanding and robotic planning and control;
2.  Specifically, the benchmark provides dense pixel-level annotations while containing more diverse physics events and being more similar to real-world videos than previous videos from simulators (e.g. [6, 29]);
3. Various baselines are evaluated and analyzed, showing to what extent the previous dynamic models understand the physical world;
4. A valuable and time-consuming human testing is provided, showing how humans perform on the new benchmark and its comparison with machine algorithms.

**Weaknesses:**

While the reviewer is positive about the benchmarks, the reviewers still have some concerns or suggestions on the project.
1. One concern is about the implementation of the baselines. The reviewer appreciates the authors provide original code links. However, according to the reviewer's understanding, baselines like RPIN and DPI are based on object/ proposal-centric representations. How do the authors handle and model the soft-body collisions for them? Will the implementation of the baselines for the new benchmark available to the public?
2. The performance comparison in Fig. 5 is easy to understanding. However, it will be hard for future researchers to copy the numbers for comparison. It will be better to provide its table versions in supplementary.
3. The DPI is based on ground-truth annotations from the simulator, which makes its comparisons with other baselines are unfair, although the reviewer knows that a fair performance comparison is not the main focus of the paper.
4. Recently,transformers [A, B] has shown strong performance for spatio-temporal reasoning. It is interesting to evaluate their performance on this new benchmark.
5. The benchmark provides dense pixel-level annotations like depth and normals. However, it seems that the baselines make no use of them. How will these visual properties help with physics understanding?
[A] Zhou, Honglu, et al. "Hopper: Multi-hop transformer for spatiotemporal reasoning." ICLR 2021.
[B] Ding D, Hill F, Santoro A, et al. "Object-based attention for spatio-temporal reasoning: Outperforming neuro-symbolic models with flexible distributed architectures", Arxiv 2021.


**Additional Feedback:**

None.

**Documentation:**

There is sufficient detail.

**Ethics:**

No.

**Relation To Prior Work:**

It has discussed how the work differs from previous contributions.

**Summary And Contributions:**

This paper studies the problem that whether machines can make future predictions for "real-world" physical events and proposes a new benchmark named Physion. It simulates photo-realistic videos with the TDW simulator[24]. The benchmark contains diverse physical scenarios, including dominoes, support, collide, contain, drop, link, roll, and drape with dense and accurate annotations like color, depth, normals ... . Various baselines are conducted and analyzed, among which DPI with object-centric graph neural networks performs the best.

---

> ### Author Response · Authors · 2021-07-09
> **Initial response to Reviewer 3: part 1/2**
>
> Thank you for your thoughtful evaluation of our manuscript and suggestions about how to improve our work. We feel that we can address four of the five weaknesses immediately and incorporate them into a revised manuscript by improving the clarity of our writing in several key sections of the paper. We also believe we can remedy the last weakness, point #4 (evaluating new models on our benchmark) within a matter of weeks, and well before the camera-ready deadline if this work is accepted for publication. Below we describe how we will do this in detail.
>
> Overall, our impression is that most of the weaknesses that you have helpfully raised are due to limitations in how we communicated our findings in the initial submission, rather than larger concerns about our approach or the value of this dataset and benchmark to the AI research community. As such, we are wondering whether there is anything else we could do or explain to increase your appraisal beyond “Marginally above acceptance threshold.”
>
> (1) Implementation of object-centric baselines and their handling of soft-body collisions. Our implementations of all models will be made public upon acceptance. Below we explain how two particular models, RPIN and DPI, could in principle handle soft-body collisions (though whether they do in practice is what our benchmark is set up to test; given that both models are ~60% accurate on the “Drape” scenario, it appears that they can handle at least some aspects of soft-body dynamics.)
>
> Both RPIN and DPI model scene dynamics with Graph Neural Networks (GNNs) that are “unrolled” for multiple time steps to simulate what will happen. In the case of RPIN, the nodes in the GNN are latent vectors (one per object) that can be *decoded* to predict the 2D centroid of each object; the components of these latent vectors, though, do not have explicit physical meaning. Because RPIN is trained to predict the future positions of each object based on interactions between these object-centric latent vectors, the components that do not represent 2D position are free to learn to represent any properties that will help with the future prediction problem. In particular, they could (but are not forced to) represent material properties of objects (such as rigidity or softness) that are relevant to the objects’ dynamics. Therefore, RPIN does not in principle require an explicit mechanism for handling rigid- or soft-body collisions separately: the dynamics of how objects interact depend on the values of their latent vector representations, which do not force objects to be “rigid” or “soft.”
>
> DPI uses a higher-resolution graph representation of scenes in which each object is encoded by a set of “particles” (graph nodes) and interactions between particles are mediated through learnable interactions. As in RPIN, the predicted particle dynamics depend on latent vector representations of each node and learnable functions that compute interactions between them, which can accommodate rigid-rigid interactions, rigid-soft interactions, and even solid-liquid interactions (as in the original work.) Moreover, the “object-centric” aspect of DPI is also implemented as a set of learnable interactions between each particle and its “object node,” with all particles that belong to the same object connected to the same root.
>
> (2) Presenting model results in table form. A table of model and human accuracy results is already in the Supplementary material (Table S1, page 19 of the full manuscript.) We will add tables that include the result of other Figure 5 subpanels.
>
> (3) How to interpret comparisons between particle-based models and vision models. You are correct that DPI (and GNS/GNS-RANSAC) receive much more information about the physical state of the simulator than do the vision models; we therefore are not surprised that they perform better on the task. However, we don’t interpret these results to mean that the vision models should be “replaced” by particle models as if they were a direct alternative.  Rather, we think of the particle models as a “strong positive control” indicating what the vision models should aspire to:  *if* a vision model *could* extract a physical representation more like that used by DPI, then it would likely more closely approach human performance. Thus, we identify this challenge as a key direction for future computer vision work. In a sense, this “recommendation” for computer vision is the most important conclusion of our benchmark.  We will make this logic more explicit in the Results and Discussion using the extra space allotted in revision; moreover, we will add visual separators to Figure 5 to indicate that the three particle-based models belong to a different category than the rest, and therefore should not be taken as “better than other vision models”, but rather as having inputs that are more useful for physical prediction.
>
> (Response continues in next comment.)

---

> > ### Author Response · Authors · 2021-07-09
> > **Initial response to Reviewer 3: part 2/2**
> >
> > (continuation of above.)
> >
> > (4) Benchmarking newer transformer-based models. We agree that it would be interesting to evaluate models with newer architectures and plan to include these (and others) in our revised manuscript.  In fact, one of the overall hopes we have for the project is that we will continually add better models going forward -- and as we do this, we plan to report results in an ongoing fashion (e.g. via leaderboard website) even once the paper itself is published.  We agree that recent work with transformers might allow for substantially improved models, and are eager to try the panoply of models within that family, including the two you referenced and a very recent self-supervised vision transformer (“DINO”) that appears to learn partly object-centric representations (https://arxiv.org/pdf/2104.14294.pdf).
> >
> > One thing that is worth noting is that we have in the current version actually already benchmarked some models that use Transformer-based visual encoders (pDeIT-MLP and pDeIT-LSTM) pretrained on ImageNet. In addition, there is a close relationship between the object-centric vision models (OP3, CSWM, and RPIN) and models like the one in Ding et al. 2021 that use an object-centric convolutional encoder and a transformer prediction module: in particular, the Graph Neural Network dynamics models used in the former are nearly equivalent to Transformers (see https://thegradient.pub/transformers-are-graph-neural-networks/) in that both architectures compute “effects” between pairs of nodes and then use these to update the node values. The fact that both GNNs and DeIT encoders were among the best of the vision models we tested motivates us to explore Transformer-based architectures more thoroughly in our revision.
> >
> > (5) Use of depth and normal annotations. You are correct that none of the baseline vision models use these geometric data; we include them in the dataset generation so that future models may either use them as input or learn geometric representations via supervision. We will include such models in our revised manuscript, such as adaptations of the model in Bear et al. 2020 (https://proceedings.neurips.cc//paper/2020/file/4324e8d0d37b110ee1a4f1633ac52df5-Paper.pdf) that incorporate these geometric features into their representations of individual objects.

---

### Official Review · Reviewer_JAHA · 2021-06-25
**Review of "Physion: Evaluating Physical Prediction from Vision in Humans and Machines"**

**Rating:** 7
**Confidence:** 4

**Strengths:**

The authors provide a resource for the critical problem of testing physical prediction and reasoning capabilities of machine learning models. The dataset is comprehensive, and encompasses a diversity of physical scenarios, and the problem formulation is elegant, distilling the task of physical prediction down to a centralized definition that is granular enough to be useful. I really like the baseline dataset that the authors provide of human physical prediction ability: the authors collect annotations from 800 human participants and compare models against human reasoning capabilities on the same dataset. That current models fall short of this baseline means that there is still ample room for our current machine learning methods to grow, and having benchmarks like these can really support this aim.

Overall, this work promises to provide a first-of-its-kind dataset for a challenging problem that has potential to advance the state-of-the-art. However, as I will explain in the weaknesses and feedback to authors section, I feel that there are some cumulative problems with the work in its current state that make it fall short of achieving this goal on a practical level.

**Weaknesses:**

I really want to like this paper. However, a major issue is that the training dataset is difficult to access because it has to be generated directly from code the authors provide, and the code is not documented well enough to explain how to generate both training datasets. But additionally, there are numerous smaller issues with the clarity of the writing, the explanation and characterization of the dataset, and plan (or lack thereof) for maintenance of the dataset, that are not fatal in isolation, but in combination result in a manuscript that is rough around the edges. These cumulative issues detract from the utility and reproducibility of the dataset,  as well as knowledge of what claims are possible with it. I provide a more detailed list of these weaknesses in the "Feedback to Author" section.

**Additional Feedback:**

1. While the authors provide mp4 and HDF5 files for their test dataset as zip files, the training dataset has to be generated from the code they provide. This limits the accessibility of the training data, especially since this problem may attract researchers from different backgrounds who may not have experience with ThreeDWorld, or have hardware limitations. I understand there may be some difficulty in making the HDF5 files for the training datasets accessible due to their large size (although the authors could consider dividing by their 8 physical scenarios), but I would like to see them make at least the mp4 files accessible. Additionally, I would prefer to see these datasets uploaded to a stable public repository (e.g. Zenodo) over a private Amazon instance, especially given there is no plan for the maintenance of these Amazon links described.

1a. How exactly do I generate the training data from the tdw_physics repository? The repository contains a README of how to generate datasets in general, which is great - but I'm not certain how I would actually reproduce the exact training data used in this manuscript. I see a "generate_original_data.sh" file, but there's no comments, and it seems to refer to a non-existent directory "target_controllers". More documentation for reproducibility purposes is essential.

2. Could the authors provide summary statistics for each of the datasets? In particular, I am interested in the class balance of contact versus no contact labels for each problem. This could just be how I sampled videos, but in the subset of mp4s I looked at for the "collide" problem, it seemed that a large majority of the scenarios contact was not made. What would be the expected accuracy if a model just predicted no contact for all inputs?

3. The authors explain their principles for producing a generated dataset with variability within each scenario, but the manuscript is unclear on what exact procedures they followed to make this happen. Examining the test data, I see that parameter sets seem to occur in "batches" - for example, I see 15 movies for the Collision scenario generated with parameters "assorted targets", "box_2", "dis_2" and "occ". Within these batches, there seem to be other parameters that are kept controlled - for example, I see the texture and friction of the flooring is consistent between all 15 movies. This is not explained in the manuscript. Importantly, since I see so many similar subsets of images in the test dataset grouped by parameter, is the test dataset images of totally different parameter sets than the training images, or a randomly held-out subset? This is critical for readers to understand exactly what makes this problem hard: I would like to know if these baselines are failing on a problem that's so hard that it can't be done in-sample, or if the authors are stacking an out-of-sample generalization problem on top of this.

4. I agree with the authors' goal of removing confounds from prior knowledge of object configurations. I can't help but notice in the videos, the authors seem to denote friction by whether the floor resembles office carpeting, or kitchen tile. Especially as this is cultural context that not all human participants would share, I can't help but wonder how a participant's ability to recognize the simulated floor as carpet/tile would have impacted their ability to solve the problem. Is there any commentary that the authors can provide on this, or statistics from the participant labels that would provide insight on this issue?

5. I have a hard time understanding the authors' explanation of the OCP readout fitting set: what does y denote here? I assumed this is P(contact), but if so, I'm unclear on how this differs from the training dataset of 2000 videos per scenario where the movies (I'm assuming t_vis in this formula is actually supposed to be in subscript) are paired with labels of P(contact)? More clarity in their explanation here is needed.

6. The authors mention adversarial examples where the dynamics are hard to capture in their methods as a case for testing if the models are learning non-causal shortcuts. I see some examples of adversarial cases in the supplementary, but what I don't see is any testing of if the models perform well on the adversarial examples.

**Clarity:**

Generally, yes - the introduction especially does a good job of outlining the motivation and utility of this dataset. However, as I point out in the feedback section, there are some issues with the explanation of how this dataset was produced (point 3), and their notation and clarity in their problem characterization (point 5).

**Correctness:**

The high-level details seem okay, and a brief browsing of the test dataset does not seem to indicate any major issues. Related to point 2 in the feedback, I'm not sure what the authors did to make sure the datasets were well-represented in contact/no-contact: was there any attempt to generate a diversity of videos for both cases, or were the videos just simulated without any prior design expectation of either? Related to point 3/2, I'm not exactly sure of how diversity of data was produced, and what the distribution of the test versus the training data is expected to look like.

**Documentation:**

The code is made open-source on a permissive license, but there is no long-term preservation plan. Additionally, the training dataset must be generated from code, but how to produce the training dataset is not sufficiently documented or explained.

**Ethics:**

Given the fully simulated nature of the dataset, and the nature of the problem, I do not foresee ethical issues.

**Relation To Prior Work:**

To my knowledge, yes, the authors provide a good overview of existing benchmarks. I am not an expert in this specific field, so I cannot comment on if there are any foundational datasets they may have missed.

**Summary And Contributions:**

Here, the authors provide a dataset of 17,200 videos of simulated physical interactions, containing 2,000 training videos and 150 test videos for 8 distinct problem scenarios. The dataset provides a clear-cut problem of predicting a final contact state between objects based upon an initial configuration and momentum. Overall, the main claims of the paper are that the dataset provides a more comprehensive challenge than existing benchmarks for physical prediction and reasoning.

---

> ### Author Response · Authors · 2021-07-09
> **Initial response to Reviewer 2: part 1/3**
>
> Thank you for your thorough reading of our work and detailed suggestions for improvement. We are glad that you think we are trying to “provide a resource for [a] critical problem” and that the underlying dataset, formulation, and human-model comparisons could be valuable. We agree that making the dataset accessible and practical to use, as well as more clearly explaining and characterizing the generation process, are critical before publication of this work. We believe that we can immediately address most if not all of your concerns, and have already begun to do so (see point-by-point response to your Feedback below.) All clarifications in our responses below will be incorporated into the final manuscript, which should be doable in the main text thanks to the extra page allotted for revision.  If there are other things we could do during the discussion period or before publication that you think would substantially improve the manuscript, please let us know.
>
> (1 + 1a): Accessibility of the training data.  At the time of submission, we only provided download links to the testing data (8 scenarios x 150 trials) as HDF5s and MP4s, and we had not fully documented the procedure for generating training data from our code using the publicly released ThreeDWorld simulator. However, it was always our intent to make this material accessible before publication. As this was one of your primary concerns, we have already made progress towards achieving this in three ways:
>
> **A. The full training dataset is now downloadable as MP4s** (https://github.com/cogtoolslab/physics-benchmarking-neurips2021/blob/master/README.md#downloading-the-training-datasets). As these files are not especially large, we have grouped all 24000 movies (8 scenarios * (2000 training + 1000 readout)) into a single archive.
>
> **B. The full training dataset is now downloadable as HDF5s** (https://github.com/cogtoolslab/physics-benchmarking-neurips2021/blob/master/README.md#downloading-the-training-datasets). While you (R2) are correct that the size of the full-scale HDF5s makes it difficult to provide them for download, none of the models we tested (and most models used in computer vision) require visual inputs other than RGB (such as depth, surface normals, optical flow.) By uploading only RGB images and object instance segmentation maps, the total size of each scenario’s training and readout datasets is between 10-50GB. This is not prohibitive for us to store and provide long term (see below) so we have now made them available.
>
> **C. Scripts for regenerating the training and readout data are provided and documented** (https://github.com/cogtoolslab/physics-benchmarking-neurips2021/blob/master/stimuli/README.md#generating-training-and-readout-data). We have updated the project page with a section on how to generate these datasets from scratch, with an explanation of how the training data distributions relate to the testing data (see below.)
>
> Long term maintenance. Thank you for prompting us to clarify our plan for long-term maintenance of our datasets. In particular, we appreciate the opportunity to articulate why we believe that Amazon S3 has a number of advantages over current alternatives (e.g., Zenodo). We thought that it would be important to clarify that we are providing the data via public S3 links, which are permanent (and permanently publicly available) URIs, as opposed to a private Amazon instance (i.e., an EC2 compute instance). S3 storage is highly redundant and extremely high-availability, and is as likely as any alternative to be maintained well into the future. Indeed, this likely explains why S3 is used so widely by large industry and commercial organizations who depend on such reliable, long-term data storage & access solutions for their very existence (e.g., Netflix). Amazon S3 does not have stringent limits on the size of stored files, a key issue since the total size of the data we provide is very substantially larger than the 50GB Zenodo limit. Amazon S3 is also highly cost-effective. For all of the training and testing data for eight scenarios, the cost to store on Amazon S3 is ~$25 / month, which we have budgeted as part of the cost of our work, and which is already covered by acknowledged funding sources that will last many years into the future. Even if there is disruption to all private/public funding sources at any point, the cost of maintenance is so low as to be easily absorbed into any single PI’s institutionally guaranteed operating budget. Thus we are confident about guaranteeing the stability and availability of this resource for at least as long as one of the PI’s labs exist (likely 25+yrs.)  Finally, please know that we are very open to migrating or duplicating our dataset on a different repository (e.g., Zenodo) in the medium-term, but upon careful reflection believe that given the above considerations, storing the data as public S3 download links is the best solution for the time being.

---

> > ### Author Response · Authors · 2021-07-09
> > **Initial response to Reviewer 2: part 2/3**
> >
> > (2) Summary statistics for the datasets. All of the testing datasets are exactly class-balanced, with 50% of the trials ending in agent-patient object contact. This was achieved by overgenerating trials that were approximately balanced (no more than 60% of trials belonging to one class) and then taking random subsamples to make the split exact. See response to point (3) below. In addition, the ground truth metadata (including outcome class label) are provided for the testing data along with human responses (https://github.com/cogtoolslab/physics-benchmarking-neurips2021#reproducing-analyses-of-human-and-modeling-behavior)
> >
> > (3) Design and generation of the datasets. While we explained our principles for generating the Physion scenarios, we did not describe our exact procedure for choosing parameter values and did not explain how the training and readout data relate to the testing data. In fact, all training and readout data were generated from the same parameter distributions as the testing data, so **there is no “out of sample generalization problem”** in testing the models trained on Physion data; the task is just very hard for the vision models we tested -- but doable for the particle-based models, suggesting the main difficulty is in mapping visual observations to a physically useful scene representation.
> >
> > The procedure we used to generate the dataset is as follows:
> >
> > For each of the eight scenarios, we first identified a handful (8-12) physical parameter distributions that (a) produced scenarios with a good mixture of contact/no-contact trials (i.e. no more than 60% of one kind), (b) in aggregate spanned a wider range of scenes related to the physical phenomenon of interest than we could produce with a single parameter distribution. As far as we can tell, the selection of “good” physical parameter distributions **cannot be easily automated:** for almost all specifications of these parameters (e.g. the range of positions, poses, sizes, and shapes of a variable number of objects), the resulting distribution of scene dynamics are boring -- nothing much happens, or all trials end with the same contact/no-contact outcome. This simply reflects the fact that the physical phenomena of interest here are very sparse under “random” dynamics: while there may be many ways to drop an object in a container, containment events rarely happen by chance. As a result of manually identifying “good” parameter distributions, the testing data come in “batches” of trials sampled from each of the configurations; the final testing sets were pruned so that they had an exact 50/50 balance of contact/no-contact trials.
> > We then used these same parameter configurations to generate the training and readout data. In the readout data, the agent object is colored red and the patient object is colored yellow, whereas in the training data these objects were randomly colored; otherwise the two types of training data come from identical distributions. Importantly, our scripts for generating training data (now provided, see response to point (1a) take into account the relative sizes of each “batch” in the testing sets, so the proportions of trials from each parameter distribution and contact/no-contact events match the testing sets.
> >
> > We will be sure to include descriptions of this procedure in the revised manuscript.
> >
> > (4) Cultural context of the Physion scenes. We share the concern that people might bring “semantic” or cultural knowledge of how different materials behave, which AI models trained from scratch lack, to the Physion task.  (Of course, this limitation is endemic to essentially all measurements of human performance characteristics in everyday behavioral contexts, and thus to any comparison of neural networks to human performance.)  Of course, our AI models are trained with access to data within the same data distribution as the test, and thus, in a sense, have some experience -- if not “cultural” experience -- of all the material types in the testing dataset. In the future, we would prefer to generate scenes in a wider range of environments (e.g. indoors, outdoors, with different types of material simulated for the ground and walls, etc.).
> >
> > To mitigate these concerns for now, we collected information about where each participant was located and some personal background (as described in the Supplement); this will allow us to analyze whether any of these variables account for some amount of variation in participant performance. In addition, we intended the Familiarization portion of the behavior experiments to help participants get a sense of how different materials behave in the simulations. Nearly all participants scored significantly above chance, suggesting that they were able to judge the material properties of scene elements well enough to make good predictions; while these data are already included in our released code, we will add a figure to the revised manuscript that makes this point explicitly.

---

> > > ### Author Response · Authors · 2021-07-09
> > > **Initial response to Reviewer 2: part 3/3**
> > >
> > > (5) Distinction between model training and readout fitting. In general, we will revise and expand section 2.4 to make the process of model training and readout fitting clearer and to better explain why we divide model evaluation into these two steps. In the original manuscript, “y” denotes the binary ground truth class label (contact or no contact), which P(contact) (i.e. the predicted class label \hat{y}) should be optimized to match. We will make the terminology consistent.
> > >
> > > Model training and readout fitting are distinct, and thus require distinct datasets. In this work, training sets were used to optimize parameters of visual encoders and/or forward dynamics models -- i.e., models that predict what is going to happen next from an initial sequence of frames. Our model training protocols included mixing all the scenarios’ training sets together, for a total of 16,000 movies each 5 seconds or longer. Importantly, though, **we did not optimize models to do the OCP task directly** during the training stage. While this is theoretically possible to do using the OCP class labels in the training datasets, it would not help us assess whether any models have or can acquire human-like representations of the physical world (see below.) Rather, we use the *readout fitting sets* to train models to do the OCP task. Specifically, we use these datasets to fit general linear models (e.g. logistic regressions) from the pretrained encoder and dynamics model features to the OCP class labels y **for a single one of the eight scenarios.** Readout fitting is really meant to play the same role as the Familiarization trials play in human experiments: a way of indicating how people or models can apply their existing representations and knowledge to do a particular psychophysics task, which may not be like any task they have previously encountered. Ideally, model readout fitting would be done only with the same 10 trials people saw during Familiarization, but here we provide two orders of magnitude more trials to make up for the fact that people in our task can follow language-level instructions, whereas the models tested here cannot.
> > >
> > > We will clarify in our revised manuscript why we do not train models directly on the OCP class labels of the training sets. In essence, it is because we are interested in *unsupervised representation learning* about the physical world, akin to unsupervised pretraining of vision or language models; we want to assess how well models trained by some other objective -- and possibly on some other dataset with more scenes and greater realism -- can *transfer* to making OCP predictions in the Physion task. This is different from supervising all of a large model’s parameters to predict the OCP class label directly on the Physion training data, which might yield models that “solve” the task (especially since the training data were drawn from the exact same distributions as the testing data) but almost certainly would yield weak, non-general representations. This would be analogous to training a large vision model with supervision to read a set of barcodes: the model might well succeed (and exceed human performance) but there is no reason to expect that it would transfer well to other visual tasks. We plan to make this logic explicit in our revision and acknowledge that it is somewhat different from other benchmarks for physical prediction, where models have usually only been tested for their ability to do a single task on a single scenario with significant supervision.
> > >
> > > Another way to think about this choice is that many of the models we test from the literature have a specifically-chosen loss function that is *part* of the model specification -- it’s not just the “architecture”  of the network that counts, but also the signal on which its representation is trained. If we had instead ripped off the original training heads of the models, and then trained just the architectures end-to-end to solve our specific red-object-hit-yellow-zone problem, that would have *not* provided a real test of what the original model creators intended. Or rather, it would just be a test of their “architecture” but not the learning signals -- and we, intentionally, explicitly seek to test the “whole models”, with their constitutive learning signals, in our benchmark.
> > >
> > > (6) Analysis of model performance on human-adversarial examples. Although discrepancies between model and human performance are already wrapped into two of our performance metrics (model-average human correlation and Cohen’s kappa), we will add analyses to the revised manuscript that explicitly compare model outputs to human responses on the “adversarial” trials. These will include examples of how various models responded on single adversarial trials as well as summary statistics of how they performed on subsets of trials as a function of human accuracy.

---

> > > > ### Comment · Reviewer_JAHA · 2021-07-12
> > > > **Reply to responses**
> > > >
> > > > Thank you to the authors for these detailed responses - I really appreciate their time and consideration thoroughly explaining the points. The major issues I raised have all been addressed: the training data is now publicly accessible. While I'm usually dicey about the stability of private repositories, I can appreciate that the large size of the training datasets is an important consideration here, and that there has been thought given to long-term maintenance of the server. Additionally, the authors have resolved my issues with the dataset balance, and I especially appreciate their explanation of the readout fitting dataset - the point about unsupervised representation learning was originally lost on me, and their explanation clarified this for me. Overall, I am satisfied with the authors' addressing of the issues I have raised, and will be raising my score during the discussion period.

---

> > > > > ### Author Response · Authors · 2021-07-14
> > > > > **Thank you!**
> > > > >
> > > > > Thank you for taking the time and care to consider both our original manuscript and our responses. This has certainly made the work stronger, clearer in its aims, and more practically useful to the AI community.

---

> ### Comment · Reviewer_JAHA · 2021-07-17
> **Updated score**
>
> I have updated my score in response to the discussion below.

---

### Official Review · Reviewer_qQpX · 2021-06-28
**Physion: Evaluating Physical Prediction from Vision in Humans and Machines**

**Rating:** 9
**Confidence:** 3
**Clarity:** The paper is clear and well written.

**Strengths:**

The dataset is very relevant for learning physical understanding of the world. The evaluation is very general, such that many types of models can be used.
The dataset is challenging. Humans cannot get perfect scores, and vision-based models are just slightly better than random.
Substantial baseline work: Several relevant baseline models are presented. This covers vision-based models (VGG + Transformer based), particle-based models (that takes the GT physical states as input), and human-evaluation (where humans are asked to estimate collision/ no collision based on small video clips)

**Weaknesses:**

The environments seems rather similar (fig. 1) (same floor, walls etc.), thus we cannot expect vision-based models to generalize to real world or other datasets, if trained on this dataset.

**Additional Feedback:**

On the github, it would be nice with some small videos to illustrate the input. This would give users a bit better intuition about the difficulty of the dataset.

#########

Final score after author feedback is unchanged. I think the dataset can be relevant for the community and the methodology well presented.

**Correctness:**

It seems to me that the dataset and baseline models are designed in a sound way.

**Documentation:**

Yes, the dataset seems reproducible. A link to their github and dataset is provided.

It would be useful on the github to provide a sample dataset (e.g. one scenario), such that users won't have to download 300 GB to get started.

**Ethics:**

No, I believe it's fine.

**Relation To Prior Work:**

Yes, the dataset is the first 3D data encompasses a wide range of physical scenarios. Furthermore, the dataset is the first large-scale dataset to compare to human level performance for action prediction.

**Summary And Contributions:**

The paper presents a new dataset with several benchmark models. The goal of the dataset is to learn physical understanding, where the physical understanding is measured by the model's/human's ability to predict collisions of basic objects based on a short video.

---

> ### Author Response · Authors · 2021-07-14
> **Initial response to Reviewer 1**
>
> Thank you for your evaluation of our work. We are glad you agree we are providing a valuable resource and initial baselining for a critical challenge in AI: understanding the physical structure and dynamics of realistic scenes. Your suggestions to provide a smaller sample dataset (not just the full 300 GB Physion testing set) and to show some videos on the landing page are well taken. We have added a GIF showing one example per scenario (https://github.com/cogtoolslab/physics-benchmarking-neurips2021) and provided links to download each of the eight testing scenarios independently, as we now do as well for the training data (https://github.com/cogtoolslab/physics-benchmarking-neurips2021#downloading-the-training-datasets).
>
> We appreciate your raising the concern that the environments in our dataset are not as diverse as the physical configurations (only two rooms are used) and that this may limit generalization of models trained on the Physion training data. We agree that models trained on Physion are unlikely to generalize to the real world, and think that training data that do allow for generalization are a desirable goal. However, we consider the core aim of the present work to be independent of and much narrower than simulation-to-reality transfer (which is an extremely broad and challenging problem in its own right): here, we mainly want to provide a **testing ground** for physical prediction models, which could be trained on whatever dataset and with whatever (self)-supervision signals the authors of those models thought essential for learning about the physical world. Since people were able to perform well on the Physion tasks from only 10 familiarization trials, we expect that an algorithm that really does understand the physical world should also be able to generalize to the combination of scenarios and environments contained in Physion.
>
> The Physion training data provided here can be used for fine-tuning or for training from scratch in cases where: (1) models are not designed to handle large-scale, realistic video data (such as OP3, CSWM, and RPIN) or (2) the required inputs and supervision signals are not available in other datasets (e.g. the particle representations used for training GRN and DPI in our work.) For a longer discussion of how we think Physion will be most useful, see our response to Reviewer 2 (point (5)) below, which will be summarized for inclusion in the revised manuscript. Of course, in an ideal world we **would** create training sets in simulation that are visually and physically diverse enough for models to generalize to the real world. Towards this end, in ongoing work we are planning to make fuller use of the capabilities of the ThreeDWorld simulator to develop such training datasets.

---

### Decision · Program_Chairs · 2021-07-26

**Decision:**

Accept

**Comment:**

The paper proposes a challenging dataset to learn physical understanding along with several benchmark models. The paper is very well-written, and the reviewers agreed that the paper should be accepted.

The physical understanding of the world is a very fundamental problem in AI research. This paper proposes a set of challenges to investigate it further. It is a very important challenge to build better model-based RL algorithms and general-purpose AI agents. The paper tests several different ML models and compares them against the human performances on those tasks. The experiments seem to be carefully designed and well-thought-out.

The authors addressed the concerns raised by the reviewers during the rebuttals quite well.